# Modified Strategies for Invasive Management of Acute Coronary Syndrome during the COVID-19 Pandemic

**DOI:** 10.3390/jcm10010024

**Published:** 2020-12-24

**Authors:** Petr Toušek, Viktor Kočka, Petr Mašek, Petr Tůma, Marek Neuberg, Markéta Nováčková, Josef Kroupa, David Bauer, Zuzana Moťovská, Petr Widimský

**Affiliations:** 1Department of Cardiology, Third Faculty of Medicine, Charles University, University Hospital Královské Vinohrady, 100 34 Prague, Czech Republic; viktor.kocka@fnkv.cz (V.K.); marketa.novackova@fnkv.cz (M.N.); josef.kroupa@fnkv.cz (J.K.); david.bauer@fnkv.cz (D.B.); zuzana.motovska@fnkv.cz (Z.M.); petr.widimsky@fnkv.cz (P.W.); 2Medtronic Czechia, Partner of INTERCARDIS Project, 190 00 Prague, Czech Republic; petr.masek@medtronic.com (P.M.); petr.tuma@fnkv.cz (P.T.); marek.neuberg@medtronic.com (M.N.)

**Keywords:** acute coronary syndrome, COVID-19, modified treatment, delays, outcome

## Abstract

The COVID-19 pandemic presents several challenges for managing patients with acute coronary syndrome (ACS). Modified treatment algorithms have been proposed for the pandemic. We assessed new algorithms proposed by The European Association of Percutaneous Cardiovascular Interventions (EAPCI) and the Acute Cardiovascular Care Association (ACCA) on patients with ACS admitted to the hospital during the COVID-19 pandemic. The COVID-19 period group (CPG) consisted of patients admitted into a high-volume centre in Prague between 1 February 2020 and 30 May 2020 (*n* = 181). The reference group (RG) included patients who had been admitted between 1 October 2018 and 31 January 2020 (*n* = 834). The proportions of patients with different types of ACS admitted before and during the pandemic did not differ significantly: in all ACS patients, KILLIP III-IV class was present in 13.9% in RG and in 9.4% of patients in CPG (*p* = 0.082). In NSTE-ACS patients, the ejection fraction was lower in the CPG than in the RG (44.7% vs. 50.7%, respectively; *p* < 0.001). The time from symptom onset to first medical contact did not differ between CPG and RG patients in the respective NSTE-ACS and STEMI groups. The time to early invasive treatment in NSTE-ACS patients and the time to reperfusion in STEMI patients were not significantly different between the RG and the CPG. In-hospital mortality did not differ between the groups in NSTE-ACS patients (odds ratio in the CPG 0.853, 95% confidence interval (CI) 0.247 to 2.951; *p* = 0.960) nor in STEMI patients (odds ratio in CPG 1.248, 95% CI 0.566 to 2.749; *p* = 0.735). Modified treatment strategies for ACS during the COVID-19 pandemic did not cause treatment delays. Hospital mortality did not differ.

## 1. Introduction

The global threat of COVID-19 and strict epidemiological containment measures have a significant impact on patients with acute coronary syndrome (ACS) in terms of contact with healthcare providers and treatment logistics after the first medical contact (FMC). The European Association of Percutaneous Cardiovascular Interventions (EAPCI) and the Acute Cardiovascular Care Association (ACCA) have proposed modified diagnostic and treatment algorithms for the COVID-19 outbreak [1].

The Czech Republic recorded some of the lowest numbers of confirmed COVID-19 cases and the lowest mortality rate in Europe during the first wave of the pandemic in spring 2020 [2]. Although the hospital system was partially re-organised, in general, there were no admission restrictions for patients with ACS.

We investigated the secondary impact of the COVID-19 pandemic on patients with ACS who were admitted to a high-volume centre in a country that was not severely affected by the initial COVID-19 wave but that implemented the algorithms proposed by the EAPCI and ACCA.

## 2. Methods

We created a prospective registry of patients with ACS admitted to the University Hospital Královské Vinohrady Cardiocentre, Prague, Czech Republic, in September 2018 (supported by the EU project INTERCARDIS–INTERventional treatment of life-threatening CARdiovascular DISeases with the cooperation of project partner Medtronic). Consecutive patients admitted with confirmed ACS were entered in the registry from 1 October 2018. ACS types were defined according to the European guidelines for acute myocardial infarction with ST elevation (STEMI) and guidelines for ACS without ST elevation (NSTE-ACS) [3,4]. Data concerning 214 parameters including clinical characteristics, angiographic, laboratory and therapeutic findings, and financial costs and hospital outcomes were obtained for all patients. The registry was approved by the local ethics committee.

Patients admitted to hospital between 1 February and 30 May 2020 were included in the COVID-19 period group (CPG). The first social media reports of COVID-19 in other countries appeared in February, and the first case of COVID-19 disease in the Czech Republic was identified on 1 March 2020, followed by strict government restrictions and school closures on 10 March 2020. Major easing of restrictions was approved by the government on 25 May 2020. During the COVID-19 period, our centre used the diagnostic and treatment strategy algorithms recommended by the EAPCI and ACCA for patients with ACS [1].

Diagnostic and therapeutic procedures for patients with stable coronary syndromes were postponed or managed according to risk stratification, which was usually conducted by phone contact. All acutely admitted patients were tested for SARS-CoV-2 immediately after admission, and the test results were received within 4–6 h (laboratory testing was performed three times daily). All patients were managed as possible COVID-19-positive until negative test results were confirmed, with healthcare workers using appropriate personal protective equipment. However, the use of invasive management was guided by clinical presentation. Patients with ongoing ischemia, STEMI, and very high-risk NSTE-ACS patients underwent immediate invasive management, and SARS-CoV-2 testing was performed afterwards. A strategy for complete revascularisation within one hospital stay was established.

The reference group (RG) included all patients entered into the registry between 1 October 2018 and 31 January 2020. The RG was divided into four four-month periods to compare the number and type of ACS within the four-month COVID-19 period. Additionally, the clinical characteristics, times to FMC and treatment, length of stay in the intensive care unit (ICU) and total hospital stay (calculated only in patients who were not transferred back to regional hospitals after the initial treatment), hospital outcomes (in-hospital patient mortality and major adverse clinical events during hospitalisation including death, re-infarction, stroke and significant bleeding (Bleeding Academic Research Consortium class > 2) and financial costs were compared between the CPG and RG. The calculation of financial costs was based on a model used nationally for diagnosis-related group costing and, for the needs of this project, was approved for the catheterisation laboratory. With the exception of expensive materials (e.g., coronary stents) and expensive drugs, which were directly registered for the patient, all expenses, including the costs of medical staff, were calculated using the cost drivers of the cost centres that provided health services, such as length of stay in bed, duration of surgery and points of procedure. The price unit calculation was based on the annual cost value of the cost centre. The costs were calculated in Czech crowns and then converted to euros according the exchange rate of the Czech National Bank on 2 July 2020.

### Statistical Analyses

The Kolmogorov–Smirnov test or Shapiro–Wilk test was used to test for normality of the data set distribution. Continuous variables with non-normal (log-normal) distributions are expressed as box plots where the central line indicates the median. Continuous variables with normal distributions are expressed as means ± standard deviations (SDs). Between-group differences were assessed using the Mann–Whitney *U*-test or Kruskal–Wallis test for non-normally distributed variables and Student’s *t*-tests for normally distributed data. The chi-square test or Fisher’s exact test was used to assess the difference between categorical variables. We also compared risk of mortality between groups expressed as odds ratio with a confidence interval (OR, CI). *p*-values < 0.05 were considered to indicate statistical significance. All statistical analyses were performed using SPSS version 26 (IBM Corp., Armonk, NY, USA). Graphical analysis was performed in Sigmaplot version 14 (Systat Software Inc., San Jose, CA, USA).

## 3. Results

### 3.1. ACS Types

The CPG consisted of 181 patients, and 834 patients were included in the RG. The proportion of acute myocardial infarction without ST-segment elevation (NSTEMI) and STEMI cases admitted in the months prior to the outbreak of COVID-19 and during the COVID-19 period were not significantly different (Figure 1). Although the number of patients with unstable angina admitted during the COVID-19 period (*n* = 23) was lower than that admitted in the previous 4-month blocks (39, 40, 37 and 46 patients, respectively), the difference was not statistically significant (*p* = 0.138).

### 3.2. Patient Characteristics

Table 1 shows the demographic and clinical characteristics of the CPG and RG patients according to ACS type (NSTE-ASC and STEMI). Age and medical history were not significantly different between groups; however, some clinical aspects were worse in CPG patients than in those in the RG. Among patients with NSTE-ACS, the ejection fraction was lower in the CPG patients than in those in the RG (44.7% vs. 50.7%, respectively; *p* < 0.001). In all ACS patients, a non-significant, higher percentage of CPG patients presented with KILLIP III-IV class at admission (13.9% vs. 9.4%; *p* = 0.082).

### 3.3. Time Intervals and Treatment Strategies

#### 3.3.1. NSTE-ACS Patients

The time from symptom onset to FMC and electrocardiogram (ECG) was less than 24 h in 49% of the CPG and in 46% of the RG (*p* = 0.588). Coronary angiography was performed in 99% of the patients in the CPG and in 99.8% of those in the RG. Of those, the procedure was performed within 24 h of admission in 49% of patients in the CPG and in 44.5% of those in the RG (*p* = 0.462).

#### 3.3.2. STEMI Patients

The time from symptom onset to FMC did not differ between the CPG and RG patients admitted to the hospital within 24 h of symptom onset (Figure 2); however, 30% of the CPG patients and 33% of the RG patients presented with subacute STEMI (time from symptom onset to FMC longer than 24 h; *p* = 0.728). The time from FMC to vessel recanalization did not differ between the RG and CPG (Figure 2).

The treatment strategies for NSTE-ACS and STEMI did not differ for the CPG and RG patients in the respective diagnostic groups (Figure 3).

### 3.4. Length of Hospital Stay

The length of stay in the ICU for patients not transferred to regional hospitals after initial treatment and the length of the total hospital stay were calculated for the patients with NSTE-ACS (Figure 4) and STEMI (Figure 5). In the NSTE-ACS patients, the ICU stay was 6.2 ± 6.6 days for the CPG and 4.7 ± 6.2 days for the RG (*p* = 0.024). The ICU stay for the STEMI patients was 4.6 ± 5.4 days for the CPG and 4.3 ± 4.3 days for the RG (*p* = 0.751). The total hospital stay for the NSTE–ACS patients was 11.1 ± 8.0 days for the CPG and 9.4 ± 9.2 days for the RG (*p* = 0.016). The total hospital stay for STEMI patients was 8.5 ± 7.0 days for the CPG and 8.0 ± 5.4 days for the RG (*p* = 0.491).

### 3.5. Hospital Outcomes

The in-hospital mortality rate for patients with NSTE-ACS was 3% in the CPG and 3.3% in the RG (odds ratio (OR) in the CPG 0.853, 95% confidence interval (CI) 0.247 to 2.951; *p* = 0.960). The in-hospital mortality rate for STEMI patients was 11.8% in the CPG and 9.6% in the RG (OR in CPG 1.248, 95% CI 0.566 to 2.749; *p* = 0.735). The major adverse event rates were not significantly different between the CPG and RG NSTE-ACS patients (8.7% vs. 6.3%, respectively; *p* = 0.389) or STEMI patients (17.9% vs. 15%, respectively; *p* = 0.491).

### 3.6. Financial Costs

The mean total hospital cost for NSTE-ACS patients was 8099 ± 6003 euros for the CPG (*p* = 0.078) and 7366 ± 6741 euros for the RG. In the STEMI patients, the total mean hospital cost was 6972 ± 5687 euros for the CPG and 6540 ± 5692 euros for the RG (*p* = 0.262).

## 4. Discussion

We reviewed an all-comers single-centre ACS registry of patient characteristics, treatment strategies, outcomes and financial costs before and during the COVID-19 pandemic when patients were managed using the EAPCI algorithms.

### 4.1. ACS Patient Characteristics

The COVID-19 outbreak was associated with a significant decline in the number of patients hospitalised for ACS in several countries, which may be explained by several patient- and system-related factors [5,6,7,8,9]. The fact that our centre did not experience a decrease in hospital admissions for patients with ACS (NSTEMI and STEMI) may be due to the relatively low involvement of the Czech Republic in the pandemic and good centre organisational protocols for ACS patients. The impact of low-intensity pandemic on hospital admissions is supported by the fact that we did not observe any patient with ACS and concomitant COVID-19 disease. Nevertheless, 8 out of the 34 patients (23.5%) admitted to the hospital with COVID-19 had slightly elevated high sensitive troponin I (less than five times higher of upper limit of range). These patients did not have other clinical symptoms of myocardial ischemia, and the troponin elevation was probably connected with myocardial injury during the inflammatory process.

### 4.2. Time Delays

In the STEMI group, the rates of primary percutaneous coronary intervention (PCI) did not differ between the CPG and RG patients and were similar to the findings of our previous large multicentre registries conducted in the Czech Republic over the last two decades [10,11,12]. The COVID-19 pandemic did not cause system-related delays in time-to-treatment intervals. Surprisingly, no differences in patient-related time delays (from symptom onset to FMC) were found between the CPG and the RG. The higher number of patients in the KILLIP III-IV class can be partially explained by the hesitation in seeking medical attention in the very early phase of ACS after the onset of symptoms. An alternative explanation may be the inability of other hospitals to provide appropriate care for critically ill patients.

The majority of NSTE-ACS patients underwent invasive procedures, and coronary angiography was performed in approximately three-quarters of the patients within the first 24 h. This is relevant because, in most cases, SARS-CoV-2 testing was performed before coronary angiography. However, patients confirmed as COVID-19 negative immediately underwent coronary angiography, which was possible because a significant reduction in elective procedures increased catheterisation laboratory availability.

Our experience differs from those reported in an EAPCI survey, which assessed the impact of the COVID-19 pandemic on interventional cardiology practice [13]. In the EAPCI survey, approximately half of the 636 respondents reported delays in reperfusion in STEMI patients (48%) and delays in early invasive treatment for NSTE-ACS patients (57%). The authors concluded that, overall, it was not possible to test patients suspected of carrying COVID-19 systematically, and the availability of personal protective equipment in the catheterisation lab was suboptimal. We did not face these limitations at our centre. Thus, it is clear that testing availability and an adequate supply of protective equipment are essential for rapid treatment of patients with ACS during the pandemic. The crucial roles of testing and organising ACS patients during the COVID-19 pandemic was also established in Lombardy for effective treatment in spring 2020. A similar algorithm for ACS patients compared to the EAPCI algorithm was developed together with centralizing cardiovascular emergencies in a regional hub-and-spokes system [14,15,16].

### 4.3. Duration of Hospital Stay and Costs

We found that among NSTE-ACS patients, the ICU stay and total hospital stay were significantly longer in the CPG than in the RG, and the total financial cost was higher in the CPG than in the RG; however, the duration of ICU and total hospital stay and total financial cost were not significantly different between the CPG and RG in STEMI patients. The differences in the patients with NSTE-ACS cannot be explained by treatment delays or different treatment strategies (the PCI and coronary artery bypass graft rates were similar in both groups). Furthermore, this cannot be explained by the second PCI performed during hospital stay to achieve complete revascularisation, which was increased during the COVID-19 period in NSTE-ACS patients as well as in STEMI patients (4% in the RG vs. 8% in the CPG between NSTE-ACS and 10 vs. 15% in STEMI patients). The most likely explanation is that the clinical condition of the NSTE-ACS patients was worse (significantly lower ejection fraction, numerically higher rate of mechanical ventilation), and NSTE-ACS patients remained in the ICU longer to stabilise their condition. Furthermore, the longer mean total hospital stay in the NSTE-ACS CPG may have been skewed by deaths of three patients after a long hospital stay (mean time from admission to death was 19 days) compared with 12 deaths in the RG (mean time from admission to death was 8 days). As a consequence, the total hospital costs were higher in the NSTE-ACS CG. These observations were not seen in the patients with STEMI.

Regarding the in-hospital mortality rate of STEMI patients, we observed higher rates of mortality than expected with respect to randomised control trials or some registries [17]. However, our in-hospital mortality rate is very similar to those of large national registries collecting data from unselected STEMI population [18,19]. Secondly, it has to be put into context with the high number of patients who died after resuscitation (in CPG, 10.3% died following out-of-hospital cardiac arrest, and 12.8% died following resuscitation but before starting PCI). The mortality rate of resuscitated patients is ten times higher than that of STEMI patients who do not require resuscitation [20]. Our registry has the highest number of resuscitated patients from all registries found in the literature. The prevalence of OHCA between STEMI patients is between 5 and 6% [21]. The in-hospital mortality rate of STEMI patients would decrease to 5.6% if OHCA STEMI patients were not included in the registry.

### 4.4. Limitations

Our study has several limitations. First, because it was a single-centre experience, our findings cannot be generalised to other European high-volume centres given that healthcare organisational strategies and local epidemiological situations differ. Second, the CPG sample was relatively small; a larger sample drawn from multiple centres would have greater statistical power. However, our aim was to conduct a quick but detailed analysis of the relevance of the proposed EAPCI algorithms for ACS management because the epidemiological situation could worsen in the future, requiring the use of this management strategy again. Collecting a large amount of data from multiple centres with sufficient quality control would take significantly longer. Third, patients admitted for ACS in February were assigned to the CPG even though the first positive case in the Czech Republic was not confirmed until 1 March 2020. However, by that time the media focus on the pandemic may have altered patient attitudes towards seeking treatment; moreover, the first steps to reorganise hospital management were taken in February. Finally, the EAPCI position statement was published online on 14 May 2020, when the epidemiological situation had stabilised. Nevertheless, our centre’s strategy, which had been established before the EAPCI publication, included all of the key EAPCI recommendations.

## 5. Conclusions

According to our centre’s experience, the COVID-19 outbreak is not associated with a decrease in the number of patients admitted with ACS when the community spread of the virus is not severe. Nevertheless, more patients were admitted to our centre with serious clinical conditions. Modified treatment strategies for patients with ACS during the first four months of the COVID-19 pandemic were not associated with treatment delays when adequate SARS-CoV-2 testing and personal protective equipment were available. Moreover, the COVID-19 situation did not have an impact on treatment strategies and hospital outcomes at our centre. However, this conclusion must be put into the context of rates of viral spread. Further detailed analyses from multicentre registries are needed to reveal more accurately the secondary impact of COVID-19 on patients with ACS.

## Figures and Tables

**Figure 1 jcm-10-00024-f001:**
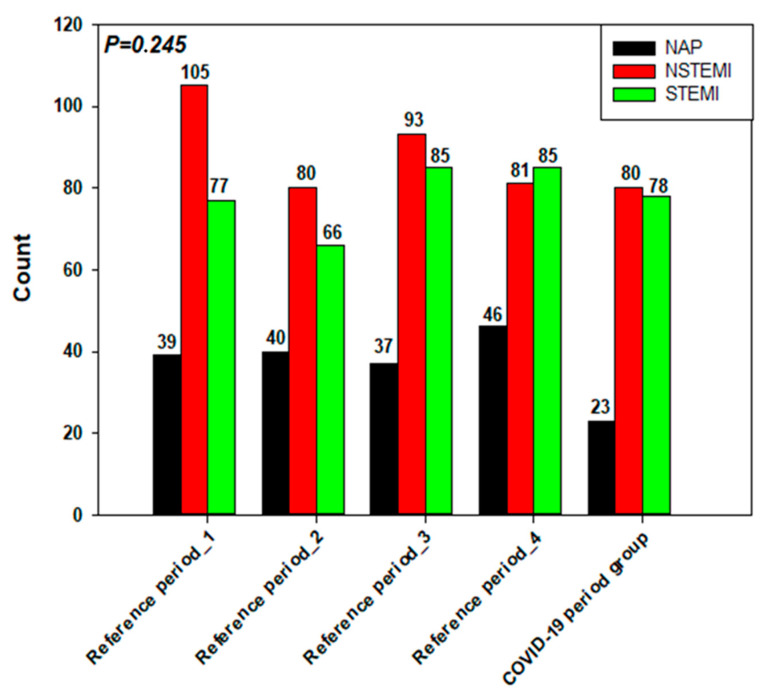
Number and types of acute coronary syndrome admitted during the four reference periods and the COVID-19 period (NAP—non-stable/unstable angina pectoris, NSTEMI—myocardial infarction with persistent ST elevation), STEMI—myocardial infarction with ST elevation).

**Figure 2 jcm-10-00024-f002:**
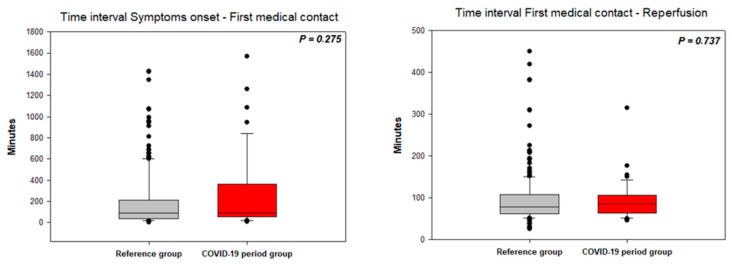
Time intervals in STEMI patients (symptoms onset—first medical contact and first medical contact to reperfusion).

**Figure 3 jcm-10-00024-f003:**
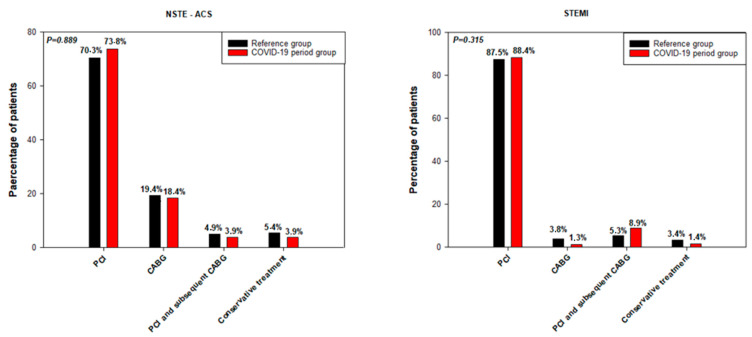
Treatment strategies for the NSTE-ACS and STEMI patients. PCI—percutaneous coronary intervention, CABG—coronary artery bypass graft.

**Figure 4 jcm-10-00024-f004:**
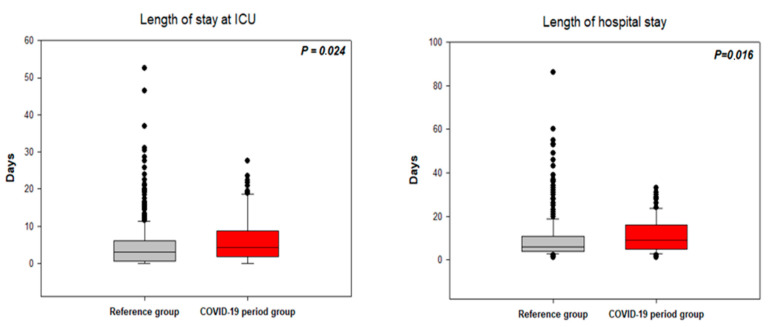
Length of stay in the intensive care unit (**left figure**) and total hospital stay (**right figure**) in patients with NSTE-ACS. ICU—Intensive care unit.

**Figure 5 jcm-10-00024-f005:**
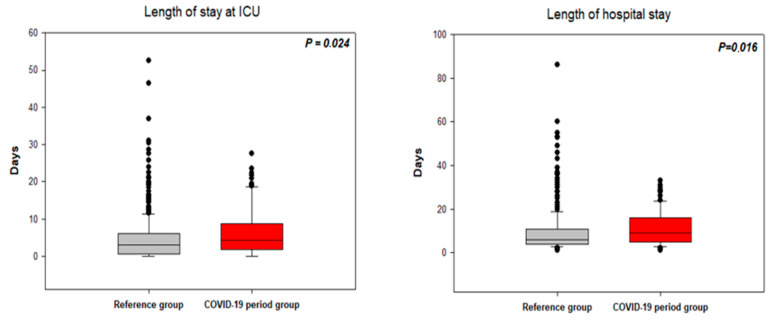
Length of stay in the intensive care unit (**left figure**) and total hospital stay (**right figure**) in patients with STEMI. ICU—Intensive care unit.

**Table 1 jcm-10-00024-t001:** Patient’s demographic and clinical characteristics

		NSTE-ACS			STEMI		
		RG	CPG	*p*-Value	RG	CPG	*p*-Value
**Age, mean (SD)**		69.5 ± 11.6	70.4 ± 10.6	0.503	65.8 ± 13.1	65.2 ± 12.4	0.9
**Male sex. *N* (%)**		381 (73.1%)	81 (78.6%)	0.270	203 (64.9%)	55 (70.5%)	0.423
**History of MI, *n* (%)**		165 (31.9%)	24 (24.2%)	0.153	38 (12.3%)	16 (21.1%)	0.05
**History of stroke, *n* (%)**		69 (13.3%)	11 (10.8%)	0.628	17 (5.5%)	7 (9.2%)	0.286
**Diabetes mellitus, *n* (%)**	**Diet**	31 (6.0%)	4 (3.9%)	0.38	19 (6.1%)	4 (5.2%)	0.366
	**PAD**	115 (22.1%)	17 (16.5%)		55 (17.7%)	11 (14.3%)	
	**Insulin therapy**	50 (9.6%)	10 (9.7%)		28 (9.0%)	3 (3.9%)	
**Hypertension, *n* (%)**		408 (78.6%)	79 (77.5%)	0.793	182 (58.9%)	46 (60.5%)	0.896
**Hyperlipidaemia, *n* (%)**		244 (47.1%)	40 (39.2%)	0.304	92 (29.8%)	29 (38.2%)	0.158
**Peripheral artery disease, *n* (%)**		73 (14.1%)	13 (12.7%)	0.875	24 (7.8%)	7 (9.2%)	0.642
**History of CABG**		71 (13.7%)	11 (10.8%)	0.523	8 (2.6%)	4 (5.3%)	0.265
**History of PCI**		152 (29.4%)	28 (27.5%)	0.722	35 (11.4%)	15 (20.0%)	0.057
**ECG rhythm**	**Sinus**	434 (83.5%)	87 (84.5%)	0.135	268 (87.6%)	67 (87%)	0.876
	**Atrial fibrillation/flutter**	56 (10.8%)	10 (9.7%)		30 (9.8%)	7 (9.1%)	
	**Pacemaker**	24 (4.6%)	2 (1.9%)		4 (1.3%)	1 (1.3%)	
	**Other**	6 (1.2%)	4 (3.9%)		4 (1.3%)	2 (2.6%)	
**KILLIP classification**	**KILLIP I**	437 (84.4%)	84 (82.4%)	0.652	244 (78.2%)	55 (71.4%)	0.324
	**KILLIP II**	43 (8.3%)	7 (6.9%)		28 (9.0%)	8 (10.4%)	
	**KILLIP III**	19 (3.7%)	6 (5.9%)		8 (2.6%)	5 (6.5%)	
	**KILLIP IV**	19 (3.7%)	5 (4.9%)		32 (10.3%)	9 (11.7%)	
**Mechanical ventilation at admission**		23(4.4%)	8 (7.7%)	0.271	21 (6.7%)	7 (9%)	0.69
**Out-of-hospital cardiac arrest**		18 (3.5%)	6 (5.8%)	0.416	25 (8%)	8(10.3%)	0.72
**Coronary angiography**	**Single vessel disease**	129 (25%)	24 (24%)	0.481	99 (32%)	23 (29.5%)	0.665
	**Multivessel disease**	379 (73.6%)	76 (76%)		210 (68%)	55 (70.5%)	
	**Left main disease**	71 (13.9%)	16 (15.7%)	0.641	22 (7.1%)	9 (11.5%)	0.241
**Ejection fraction**		50.7 ± 11.1	44.7 ± 16.2	0.001	43.1 ± 10.8	42.5 ± 12.3	0.994

NSTE-ACS—acute coronary syndromes without ST elevation, STEMI—myocardial infarction with ST elevation, RG—reference group, CPG—COVID-19 reference group, MI—myocardial infarction, PAD—peroral antidiabetic treatment, CABG—coronary artery bypass graft, PCI—percutaneous coronary intervention.

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
