# Peer review of "Modified Strategies for Invasive Management of Acute Coronary Syndrome during the COVID-19 Pandemic"

_jcm, 2020, doi:10.3390/jcm10010024_

Round 1
Reviewer 1 Report
The authors have made appropriate edits , considering the data that they have, to my previous comments
Author Response
Thank you very much for accepting our changes. There are no other comments from the reviewer.
Reviewer 2 Report
The manuscript by Petr Toušek et al aims assess new algorithms proposed by The European Association of Percutaneous Cardiovascular Interventions (EAPCI) and the Acute Cardiovascular Care Association (ACCA) on patients with ACS admitted to hospital during the COVID-19 pandemic. Authors conclude that themodified treatment strategies for ACS during the COVID-19 pandemic did not cause treatment delays in ACS patients. Of note, Czech Republic was one of the European country less involved in COVID-19 pandemic during spring 2020, so results obtained cannot be generalized to other countries.
The manuscript is well written and statistical analysis is appropriate. Abbreviations must be explained in every figure or table caption in order to be more intelligible.
Of note, patients admitted for ACS in February were assigned to the CPG even though the first positive case in the Czech Republic was not confirmed until 1 March, 2020, as authors declare in limitations. However, if the first steps to reorganize hospital management were taken in February, the results should have not been influenced by this issue.
Author Response
The manuscript is well written and statistical analysis is appropriate. Abbreviations must be explained in every figure or table caption in order to be more intelligible.
Thank you for this comment. We added description of abreviations to all figures and to the table.
Of note, patients admitted for ACS in February were assigned to the CPG even though the first positive case in the Czech Republic was not confirmed until 1 March, 2020, as authors declare in limitations. However, if the first steps to reorganize hospital management were taken in February, the results should have not been influenced by this issue.
Thank you for mentioning this issue. We are fully aware that including February into the analysis is one of the limitation and we are discussing this issue in the limitation section. On the other hand, the first steps of reorganizations were taken in February as reviewer stated. Futhermore, our goal was to analyse the patients behaviour (interval onset symptoms – first medical contact) during the global threath of Covid-19. There was a huge media information campaign about COVID-19 disease.
Reviewer 3 Report
This is an analysis of the outcomes of ACS patients treated at a single large centre, with a focus on understanding the impact of COVID-19. The authors decided to present their data as a "COVID group" versus a "reference group" ie pre-COVID-19.
However, the COVID group includes patients admitted from February 1st whereas the first case of COVID-19 was not seen in the country until 1st March. Whilst I appreciate that the authors wanted to see if there were any changes in the time taken for patients to call for help, I do not think that they can justify starting the data collection for the COVID group prior to there being a single case in the country. In their current analysis, one quarter of their "COVID group" patients were admitted when there wasn't any COVID in their country / hospital, so I think that the analysis should be re-done.
What I do agree on though is that in the absence of a large number of positive patients admitted with the ACS pathways, they have shown that outcomes are no different to the reference group. This is important as it demonstrates that even during a pandemic, if infection rates are low, it is still feasible to follow the "usual pathways" of best clinical practice using appropriate precautions, PPE etc.
It is very surprising that the authors did not see/include a single patient who was positive for COVID-19. This suggests that those admitted with COVID-19 were not tested for troponin despite this being a marker of prognosis. Can the authors please describe in more detail their pathway / diagnostic criteria for NSTEMI. Is it possible that those positive for COVID-19 and with high troponin levels were not accepted onto the cardiology department even if they had ECG abnormalities?
How many patients were admitted to their hospital due to COVID-19 in the same period?
The authors seem to indicate that there is a difference in Killip class between the 2 groups however this is not the case - there was no statistically significant difference.
There are some minor adjustments needed to the English.
Please change the word medial to media in line 253.
Author Response
Thank you very much for the review and important comments. We have answered all points bellow and did some changes in the text of manuscript according suggestions.
This is an analysis of the outcomes of ACS patients treated at a single large centre, with a focus on understanding the impact of COVID-19. The authors decided to present their data as a "COVID group" versus a "reference group" ie pre-COVID-19. However, the COVID group includes patients admitted from February 1st whereas the first case of COVID-19 was not seen in the country until 1st March. Whilst I appreciate that the authors wanted to see if there were any changes in the time taken for patients to call for help, I do not think that they can justify starting the data collection for the COVID group prior to there being a single case in the country. In their current analysis, one quarter of their "COVID group" patients were admitted when there wasn't any COVID in their country / hospital, so I think that the analysis should be re-done.
We compleatly agree wth the reviewer that including February into the the Covide-19 period in one of the limitation. This was mentioned in the limitation section. On the other hand, the first steps of reorganizations were taken in February. Futhermore, our goal was to analyse the patients behaviour (interval onset symptoms – first medical contact) during the global threath of Covid-19 that was certainly present in February. In our analysis we did not observed significant changes in this intervenl during February nor in other months of COVID-19 period. We allow to keep this analysis as was planned in methodology of the study.
What I do agree on though is that in the absence of a large number of positive patients admitted with the ACS pathways, they have shown that outcomes are no different to the reference group. This is important as it demonstrates that even during a pandemic, if infection rates are low, it is still feasible to follow the "usual pathways" of best clinical practice using appropriate precautions, PPE etc.
Thank you for this comment.
It is very surprising that the authors did not see/include a single patient who was positive for COVID-19. This suggests that those admitted with COVID-19 were not tested for troponin despite this being a marker of prognosis. Can the authors please describe in more detail their pathway / diagnostic criteria for NSTEMI. Is it possible that those positive for COVID-19 and with high troponin levels were not accepted onto the cardiology department even if they had ECG abnormalities?
Thank you for this comment. We have used the same diagnostic criteria for NSTEMI as described by ESC guidelines. Rise/fall of troponin plus at least one of the signs for myocardial ischemia (clinical symptoms/ECG/other imaging showing ischemia or necrosis).
How many patients were admitted to their hospital due to COVID-19 in the same period?
There were 34 patients admitted due to COVID-19 disease to the hospital during the period. Out of them, 8 patients (23,5%) have increased high sensitivity Troponin I (hsTnT). However the elevation did not exceeded 5 times upper range of limit for hsTnT, patients did not have clinical or imaging signs suggesting myocardial ischemia. Thus, we think, that these 8 patients cannot be regarded as patients with acute coronary syndrome and increased troponin levels were associated with mild myocardial injury during the inflammatory process in COVID-19 disease. This is now discussed int the text.
The authors seem to indicate that there is a difference in Killip class between the 2 groups however this is not the case - there was no statistically significant difference.
Thank you for this comment. We have changed the expresion in abstract that does not make the impression of significant difference in KILLIP III-IV class between both groups.
There are some minor adjustments needed to the English.
The text have been controlled by native English speaker.
Please change the word medial to media in line 253.
The word has been changed.
This manuscript is a resubmission of an earlier submission. The following is a list of the peer review reports and author responses from that submission.
Round 1
Reviewer 1 Report
In this study, Tousek et al provide a well-written manuscript describing the clinical profil and in-hospital outcome and cost of patients presenting with acute coronary syndrome during the COVID pandemic, compared to a historical cohort. They report no major significant difference between the two cohorts with respect to delay from symptom onset to PCI or in-hospital outcomes.
Overall these data are interesting, and provide reassuring results as to the management of CAD during the COVID pandemic in Czech Republic. However, as mentioned by the authors, the Czech Republic, was fortunately not presenting with a significant surge of CVODI cases during the period of interest. Conversely, there has been several reports, notably from Italy or France, which reported a reduction of hospitalization for cardiovascular disease with a surge of extra-hospital cardiac arrest during the pandemic(1–3). To the credit of the author, this limitation is appropriately acknowledged several times in the manuscript by the authors.
I have other more specific comments:
-First and foremost: It would have been of interest to provide the actual number of patients that were eventually diagnosed with concomitant COVID.
- The time from symptom onset to FMC and electrocardiogram (ECG) was less than 24 h in 49% of the CG and in 46% of the RG (p = 0.588) => do you have the actual delay? It could be significantly different between the two groups, while remaining overall below 24h. I would suggest providing the mean delay between the group, fro both STEMI and NSTEMI if possible
- You start by stating in the abstract that you aimed at evaluating modified strategy to deal with patient presenting ACS, but you don’t actually describe them in the abstract. Consider at least mentioning the European Association of 45 Percutaneous Cardiovascular Interventions (EAPCI) and the Acute Cardiovascular Care Association 46 (ACCA) have proposed modified diagnostic and treatment algorithms in the abstract
- A strategy for complete revascularization within one hospital stay was established.=> was this part of the EAPCI/ACCA guidelines or a pragmatic decision?
Author Response
Thank to reviewer for his important comment, we have changed our manuscript accordigly in the majority of requested points.
Reviewers comments and authors answers:
First and foremost: It would have been of interest to provide the actual number of patients that were eventually diagnosed with concomitant COVID.
Thank to reviewer for this important point. We did not diagnosed any patient with acute coronary syndrome and concomitant COVID-19 disease during the COVID-19 period. We have added this information into to discussion section. This information is in context of low intensity of pandemic during the spring months of 2020.
- The time from symptom onset to FMC and electrocardiogram (ECG) was less than 24 h in 49% of the CG and in 46% of the RG (p = 0.588) => do you have the actual delay? It could be significantly different between the two groups, while remaining overall below 24h. I would suggest providing the mean delay between the group, fro both STEMI and NSTEMI if possible.
Unfortunately we did not measured the delay in NSTE-ACS patients between FMC and ECG in minutes like in STEMI patients. We have just assigned them into different time period (0-12, 12-24, 24-48 and more than 48hours). It is much more difficult the measure the exact time when patients are referred from other hospital and many times the informations are missing. We used the same strategy for the time delay measurement like in NSTE-ACS guidelines.
- You start by stating in the abstract that you aimed at evaluating modified strategy to deal with patient presenting ACS, but you don’t actually describe them in the abstract. Consider at least mentioning the European Association of 45 Percutaneous Cardiovascular Interventions (EAPCI) and the Acute Cardiovascular Care Association 46 (ACCA) have proposed modified diagnostic and treatment algorithms in the abstract
I totaly agree with the reviewer. Unfortunately, instructions for authors allows us only 200 words in abstract and we had to shorten the word count. We kept most important information in abstract and we did not have more space to mention EAPCI and ACCA proposed algorithm in abstract. Now, this information was added into the abstract ad we hope it will be acceptable for the editorial office.
- A strategy for complete revascularization within one hospital stay was established.=> was this part of the EAPCI/ACCA guidelines or a pragmatic decision?
This was a pragmatic decision and it was also part of EPACI/ACCA algorithm.
Reviewer 2 Report
Modified strategies for invasive management of acute coronary syndrome during the COVID-19 pandemic.
Petr Toušek and al reported Czech Republic experience with ACS management during COVID 19 pandemic. One major reservation is that this country had some of the lowest numbers of confirmed COVID-19 cases and thus, raises the issue of the generalisation of the results…
Otherwise, the manuscript is clear and well-written.
Minor comments
- It is not clear if patients in “COVID-19 group” had COVID-19? It is disturbing to name the ”COVID-19 group" while these patients do not have necessarely COVID-19. The group name should be changed.
- Line 179 : please add the following reference: Hauguel-Moreau M, Pillière R, Prati G, Beaune S, Loeb T, Lannou S, Mallet S, Mustafic H, Bégué C, Dubourg O, Mansencal N. Impact of Coronavirus Disease 2019 outbreak on acute coronary syndrome admissions: four weeks to reverse the trend. J Thromb Thrombolysis. 2020 Jun 29:1–2. doi: 10.1007/s11239-020-02201-9. Epub ahead of print. PMID: 32601849; PMCID: PMC7323878.
- Line 250: please remove this part of the sentence which is insulting for other centre: “and when the centre is well organised.”
Author Response
Thank to reviewer for his comments we have acceted all suggested changes:
It is not clear if patients in “COVID-19 group” had COVID-19? It is disturbing to name the ”COVID-19 group" while these patients do not have necessarely COVID-19. The group name should be changed.
We agree with the reviewer that using the term of COVID-19 group can be confusing. We have changed the term into COVID-19 period group (CPG).
Line 179 : please add the following reference: Hauguel-Moreau M, Pillière R, Prati G, Beaune S, Loeb T, Lannou S, Mallet S, Mustafic H, Bégué C, Dubourg O, Mansencal N. Impact of Coronavirus Disease 2019 outbreak on acute coronary syndrome admissions: four weeks to reverse the trend. J Thromb Thrombolysis. 2020 Jun 29:1–2. doi: 10.1007/s11239-020-02201-9. Epub ahead of print. PMID: 32601849; PMCID: PMC7323878.
This reference was added.
Line 250: please remove this part of the sentence which is insulting for other centre: “and when the centre is well organised.”
This sentence was removed as requested.